# Screening of Potential α-Glucosidase Inhibitors from the Roots and Rhizomes of *Panax Ginseng* by Affinity Ultrafiltration Screening Coupled with UPLC-ESI-Orbitrap-MS Method

**DOI:** 10.3390/molecules28052069

**Published:** 2023-02-22

**Authors:** Hong-Ping Wang, Chun-Lan Fan, Zhao-Zhou Lin, Qiong Yin, Chen Zhao, Ping Peng, Run Zhang, Zi-Jian Wang, Jing Du, Zhi-Bin Wang

**Affiliations:** 1Scientific Research Institute of Beijing Tongrentang Co., Ltd., Beijing 100011, China; 2Beijing Tongrentang Technology Development Co., Ltd., Beijing 100079, China; 3Beijing Zhongyan Tongrentang Pharmaceutical R & D Co., Ltd., Beijing 100000, China

**Keywords:** *panax ginseng*, α-Glucosidase inhibitors, affinity ultrafiltration screening, LC-MS

## Abstract

*Panax ginseng* was a traditional Chinese medicine with various pharmacological activities and one of its important activities was hypoglycemic activity; therefore, *panax ginseng* has been used in China as an adjuvant in the treatment of diabetes mellitus. In vivo and in vitro tests have revealed that ginsenosides, which are derived from the roots and rhizomes of *panax ginseng* have anti-diabetic effects and produce different hypoglycemic mechanisms by acting on some specific molecular targets, such as SGLT1, GLP-1, GLUTs, AMPK, and FOXO1. α-Glucosidase is another important hypoglycemic molecular target, and its inhibitors can inhibit the activity of α-Glucosidase so as to delay the absorption of dietary carbohydrates and finally reduce postprandial blood sugar. However, whether ginsenosides have the hypoglycemic mechanism of inhibiting α-Glucosidase activity, and which ginsenosides exactly attribute to the inhibitory effect as well as the inhibition degree are not clear, which needs to be addressed and systematically studied. To solve this problem, affinity ultrafiltration screening coupled with UPLC-ESI-Orbitrap-MS technology was used to systematically select α-Glucosidase inhibitors from *panax ginseng*. The ligands were selected through our established effective data process workflow based on systematically analyzing all compounds in the sample and control specimens. As a result, a total of 24 α-Glucosidase inhibitors were selected from *panax ginseng*, and it was the first time that ginsenosides were systematically studied for the inhibition of α-Glucosidase. Meanwhile, our study revealed that inhibiting α-Glucosidase activity probably was another important mechanism for ginsenosides treating diabetes mellitus. In addition, our established data process workflow can be used to select the active ligands from other natural products using affinity ultrafiltration screening.

## 1. Introduction

*Panax ginseng*, which is a traditional medicinal and edible plant, has been widely used in China and Asia for thousands of years. Modern research has proved that *panax ginseng* has various pharmacological activities, and as well as having anti-tumor [1], anti-aging [2], anti-oxidation [3], anti-fatigue [4], improvement of immune function, intelligence as well as learning ability [5,6], a protective effect on cerebral ischemia, liver as well as kidneys [7,8,9], and regulation of central nervous system [10], *panax ginseng* also has the effect of decreasing blood sugar and has been used in China as an adjuvant in the treatment of diabetes mellitus. Accumulating evidence has shown that ginsenosides, which are extracted from *panax ginseng*, exert anti-diabetic effects [11,12,13,14,15]. In vivo and in vitro tests revealed that ginsenosides have anti-diabetic effects mainly because they act on some specific molecular targets. For example, ginsenoside Rg_1_, Rg_3_, F_2_, compound K, and Rh_2_ could effectively reduce intestinal glucose uptake through the regulation of sodium-glucose cotransporters 1 (SGLT1) gene expression [16,17]; ginsenoside Rg_3_ reduces blood glucose and increases plasma glucagon-like peptide-1(GLP-1) and plasma insulin through the improvement of insulin resistance, lipid metabolism, energy metabolism, and gut flora metabolism [11]; ginsenoside Rb_1_ can promote the translocation of glucose transporter to increase glucose uptake in adipocytes, and this reduces fasting glucose through recovery in the expression of glucose transporters (GLUTs) and the phosphorylation of Akt in the adipose tissue of db/db mice [18]; ginsenoside Rg_1_, Rb_3,_ and compound K can reduce gluconeogenesis through increase AMP-activated protein kinase (AMPK) expression and decreased Forkhead transcription factor 1 (FOXO1) activity [19,20,21], etc. All these modern research results demonstrate that ginsenosides can produce different hypoglycemic mechanisms by acting on different molecule targets. However, besides the above-mentioned pharmacological mechanisms of the anti-diabetic effects of drugs, inhibiting the activity of α-Glucosidase so as to delay the absorption of dietary carbohydrates and finally reduce postprandial blood sugar is another important hypoglycemic mechanism. For ginsenosides, the anti-diabetic effects, whether they have the mechanism of inhibiting α-Glucosidase activity, and which ginsenosides exactly attribute to the inhibitory effect as well as the inhibition degree are not clear and need to be addressed and systematically studied. 

In a previous study, the main approach for identifying active ingredients from the roots of *panax ginseng* was separately testing their biological activity after chemical separation one by one [22]. However, the experiment period is long with a heavy workload, and even some active ingredients with trace amounts are lost during separation. Moreover, it is impossible to achieve high-throughput screening. In order to achieve rapid screening and identification of α-glucosidase inhibitors from natural products, some researchers developed a novel at-line nanofractionation screening platform, in which a time-course bioassay based on high-density well-plates was performed in parallel with high-resolution mass spectrometry (MS), providing a straightforward and rapid procedure to simultaneously obtain chemical and biological information of active compounds, and with this approach, the efficiency and quality of screening can be increased [23]. However, in our study, another much more useful technology for screening α-glucosidase inhibitors, affinity ultrafiltration screening-mass spectrometry, which combined affinity ultrafiltration screening technology with liquid chromatogram-mass spectrometry (LC-MS) technology, was employed. Due to it having the advantage of making the screening of active components from natural products more convenient and effective, it has played an increasingly vital role in early drug discovery [24,25,26,27]. Through affinity ultrafiltration screening technology, the small molecule ligands bound to biological macromolecules can be obtained, and through LC-MS technology, especially through ultra-performance liquid chromatography (UPLC) coupled with ESI-Orbitrap-MS technology which can provide high-resolution MS spectra, the structures of the selected ligands can be rapid analysis and characterization. However, during the selection of ligands from dissociation solution obtained through affinity ultrafiltration screening, some researchers visually compared chromatographic peak intensity on liquid chromatograms of the sample group containing target protein with the control group containing denatured/unadded target protein, and the compounds with higher intensities in the sample group were selected as the ligands [24,25,26,27]. However, when peaks contain many overlapping signals, this approach will lead to false positive or false negative results. Under these circumstances, an effective data process workflow for ligands selection should be established.

Based on the above problems that need to be solved urgently, we used affinity ultrafiltration screening coupled with UPLC-ESI-Orbitrap-MS technology to systematically select α-Glucosidase inhibitors from *panax ginseng*, in which the major components were ginsenosides, and meanwhile established an effective data process workflow to systematically select the small molecule ligands instead of previous visual comparison of chromatographic peak intensity on liquid chromatograms of the sample and the control specimens. Here, through our integrated affinity ultrafiltration screening-MS and established data process workflow, a total of 24 active ingredients were selected from *panax ginseng*, including 14 known ginsenosides and 10 potential new ginsenosides. Our study first systematically selected and characterized the α-Glucosidase inhibitors from *panax ginseng* and revealed that inhibiting α-Glucosidase activity probably was another important mechanism for the hypoglycemic effect of ginsenosides, which will improve cognition in people undergoing ginsenosides treatment of diabetes mellitus.

## 2. Results and Discussion

### 2.1. Selection of α-Glucosidase Inhibitors by Affinity Ultrafiltration Screening–LC-UPLC-ESI-Orbitrap-MS

The designed process of selection of α-Glucosidase inhibitors is briefly introduced as follows: when the extract of *panax ginseng* was incubated with α-Glucosidase, active ligands could bind to the active site of α-Glucosidase forming receptor–ligands complexes, and unbound small molecules were free, which could be separated from receptor–ligands complexes by using an ultrafiltration membrane. Then, the receptor–ligands complexes were disrupted by the addition of dissociation agent, and the released ligands were analyzed by UPLC-ESI-Orbitrap-MS analysis. The full scan data was analyzed by our established data process workflow based on systematic analysis of all compounds in the specimens to select the active ingredients, and the ingredient with the peak area ratio (PAR) value, which was defined as the ratio of peak area of a compound detected in the sample group to that detected in the control group, >1 and *p* < 0.05 (*n* = 4) was selected as the ligand. 

Figure 1 shows the total ion chromatograms of *panax ginseng* extract, sample, and control specimens. From their total ion chromatograms, we found that there were many overlapping signals due to ginsenosides in *panax ginseng* owning similar chemical structures and polarity, leading to their poor separation in the reversed chromatographic column. Even in the optimized chromatographic conditions, it is still difficult to achieve their complete separation. Therefore, directly comparing the intensity of each peak in their total ion chromatograms between the sample and the control groups can lead to false positive or false negative results. In our experiment, we established an effective workflow to systematically analyze all compounds in the sample specimens, and compounds not only with high abundance but also with lower intensity even covered by others were analyzed. Each compound of PAR value and its mean PAR value were calculated according to the obtained peak areas yielded by the compounds-extracting workflow. A two-tailed t-test was used to calculate the significant difference (*p* value) in the peak area of each compound between the sample and the control group. The compounds, whose mean PAR values >1 and *p* < 0.05 (*n* = 4), were selected as the potential α-Glucosidase inhibitors, and as a result, 24 ligands (shown in Table 1 and Appendix A) including 14 known ginsenosides (R1–R14) and 10 unknown ginsenosides (R15–R24) were successfully selected, and the data of affinity ultrafiltration screening was shown in Appendix A.

From Table 1, we found that the PAR values of the selected were all >1, meaning the intensities of these compounds in the sample group were higher than those in the corresponding control group. In order to confirm that there was indeed a difference in peak intensity between the sample and the control group for the selected ligands, we separately extracted their ion chromatograms in the sample and control specimens, and we found all of the selected ligands exhibiting higher intensities in the sample group, meaning our established data process workflow was reliable and effective. Figure 2 showed the PAR values of some representative ligands. The precursor ions of all the selected ligands were then used to perform targeted MS/MS analysis for structure identification. Although we could characterize the selected ligands based on their high-resolution MS and fragmentation ions produced by targeted MS/MS analysis, unambiguous structural elucidation requires strict confirmation with reference standards. To this end, we obtained the 14 reference standards (R1–R14) (shown in Figure 3) through commercial sources. The identification of these ligands is based on matching both retention time deviation (<0.1 min) and accurate mass deviation (<5 ppm) with the corresponding reference standards. The unknown ligands R15–R24 were elucidated according to the diagnostic ions and fragmentation pathways of ginsenosides, and we found that most of the unknown ligands were the isomers of the known ligands. For example, R15 and R18 showed the same precursor ions at *m*/*z* 955.4904 with a mass deviation of 0.10 ppm indicating their molecular formula was C_48_H_76_O_19_. In their MS/MS spectra, their diagnostic ions were observed at *m*/*z* 455.3534 suggesting that they were oleanolic-type ginsenosides. The fragmentation ions at *m*/*z* 793.4376, 731.4376, 613.3737, 569.3847, and 455.3534 were the same as those of ginsenoside Ro (R8). Thus, R15 and R18 were separately deduced as ginsenoside Ro isomers. Similarly, R16 and R22 were separately deduced as ginsenoside Ra_1_ isomer or ginsenoside Ra_2_ isomer due to their fragmentation ions being the same as those of ginsenoside Ra_1_ or ginsenoside Ra_2_. R24 was deduced as a quinquenoside R_1_ isomer due to its fragmentation ions were the same as those of quinquenoside R_1_. In the MS/MS spectra of R17 and R21, after the loss of Ac (42 Da), their remaining fragmentation ions were the same as those of ginsenoside Rd; thus, R17 and R21 were separately deduced as acetyl-ginsenoside Rd. In the same way, R19 was deduced as (*E*)-but-2-enoyl ginsenoside Rd because after the loss of (*E*)-but-2-enoyl (68 Da), the remaining fragmentation ions were the same as those of ginsenoside Rd. The precursor ion of R20 was observed at *m*/*z* 943.5251 with a mass deviation of −1.59 ppm indicating its molecular formula was C_48_H_80_O_18_. In its MS/MS spectra, its aglycone ion was obtained at *m*/*z* 457.3699, which was 18 Da less than the aglycone of protopanaxatriol (*m*/*z* 475 in negative-ion mode); thus, we considered the aglycone ion *m*/*z* 457.3699 was dehydrated-protopanaxatriol. The fragmentation ions of *m*/*z* 781.4766, 619.4214, and 457.3699 suggested that Glc, Glc, Glc were successively eliminated from the precursor ion. Thus, R20 was deduced as dehydrated-protopanaxatriol + 3Glc. The precursor ion of R23 was observed at *m*/*z* 1105.5779 with a mass deviation of −1.45 ppm, suggesting its molecular formula was C_54_H_90_O_23_. In its MS/MS spectra, its aglycone ion was obtained at *m*/*z* 457.3698, which was formed by successive losses of 4Glc from the precursor ion *m*/*z* 1105.5779. Thus, R23 was deduced as dehydrated-protopanaxatriol + 4Glc. The fragmentation ions of all 24 ligands are shown in Table 1.

As we know, there are mainly three types of ginsenosides in *panax ginseng*, including protopanaxadiol-, protopanaxatriol-, and oleanolic acid-type. Among the selected ligands, seven ginsenosides including R1, R4–R6, R8, R15, and R18 belong to oleanolic acid-type, whereas thirteen ginsenosides including R2, R7, R10–R14, R16, R17, R19, R21, R22 and R24 belong to protopanaxadiol-type. However, only four ginsenosides, including R3, R9, R20, and R23 are protopanaxatriol-type. Thus, it can be seen that protopanaxadiol-type ginsenosides are the main α-Glucosidase inhibitors, and oleanolic acid-type ginsenosides come second. Nevertheless, protopanaxatriol-type ginsenosides which are one of the most important ginsenosides in *panax ginseng* own the least number of ligands with much lower abundance. 

It is worth noting that through our established data process workflow, much more ligands were selected, and not only were the known compounds with higher amounts (R1–R14) selected and characterized as α-Glucosidase inhibitors but also some unknown compounds (R15–R24) with lower abundance covered by other compounds, and these unknown compounds were probably potential new compounds.

### 2.2. Molecular Docking of α-Glucosidase and Ligands

In order to predict the affinity between the selected ligands and α-Glucosidase, the selected ligands were docked with α-Glucosidase. The compounds (R1–R14) owning the definite structures were downloaded from PubChem and their 3D structures were docked with pre-processed α-Glucosidase. Generally, when the protein–ligand interaction was analyzed using AutoDock vina (version 1.5.6), affinity ≤ −7 kcal/mol indicated the compounds had strong binding with the target protein. After molecular docking with α-Glucosidase, the affinities of R1–R14 were all in the range of −7.2–−9.0 kcal/mol, indicating they all had a strong affinity with α-Glucosidase. In addition, acarbose, which was widely used as an α-Glucosidase inhibitor, was also docked with the pre-processed α-Glucosidase and its affinity was −7.1 kcal/mol. From another aspect, the results mentioned above verified the reliability of the affinity ultrafiltration screening. However, R15–R24 could not be docked with α-Glucosidase due to the fact that they were probably new compounds, and their definite structures were unknown yet.

### 2.3. α-Glucosidase Inhibitory Activity of Ligands

To further verify the α-Glucosidase inhibitory activity of the selected ligands, in vitro enzyme inhibition assay was performed. Due to only the reference standards of R1–R14 being commercially available, these fourteen compounds were performed in vitro enzyme inhibition assay. However, due to R2 and R7 being insoluble in a 0.1 M phosphate buffer (pH 6.8) even though co-solvent DMSO (2%) was added, their in vitro enzyme inhibition assay was not performed. The remaining twelve compounds were easily soluble in a 0.1 M phosphate buffer (pH 6.8) and their α-Glucosidase inhibitory activities were tested. The results are shown in Table 2. From Table 2, we found that oleanolic acid-type ginsenoside zingibroside R_1_ (R1) exhibited much stronger α-Glucosidase inhibitory activity with the IC_50_ value of 3.61 mM, even superior to the positive control acarbose (IC_50_ value of 5.25 mM). Nevertheless, the other selected oleanolic acid-type ginsenosides displayed weaker α-Glucosidase inhibitory activities than acarbose, such as pseudoginsenoside-RT_1_ (R4) and chikusetsusaponin Iva (R6) with the IC_50_ values of 39.30 mM and 17.33 mM, respectively, whereas chikusetsusaponin IV (R5) and ginsenoside Ro (R8) separately exhibited 16.20% and 20.23% inhibition rate at 40 mM. For the tested protopanaxadiol-type ginsenosides, all of them displayed lower α-Glucosidase inhibitory activities, and except for ginsenoside Rc (R14) was with the IC_50_ value 36.83 mM, the inhibition rate of ginsenoside Ra_1_ (R10), as well as ginsenoside Ra_2_ (R11), were <30% at 40 mM whereas the inhibition rate of quinquenoside R_1_ (R12), as well as ginsenoside Ra_3_ (R13), were <20% at 24 mM. For the tested protopanaxatriol-type ginsenosides, ginsenoside F_4_ (R9) displayed stronger α-Glucosidase inhibitory activity than ginsenoside Rg_6_ (R3). From Figure 3, we found that the difference in their structures is the double bond position at C21, and ginsenoside F_4_ (R9) owns a non-terminal double bond whereas ginsenoside Rg_6_ (R3) owns a terminal double bond. This implied that the double bond position of ginsenosides could affect the activation degree of ginsenosides, and the non-terminal double bond was better than the terminal double bond.

## 3. Materials and Methods

### 3.1. Samples, Reference Standards, and Reagents

α-Glucosidase from *Saccharomyces cerevisiae* was purchased from Sigma (Enzyme Commission number: 3.2.1.20, 100 units, St. Louis, MO, USA), and *p*-Nitrophenyl-α-d-glucopyranoside (pNPG) was obtained from Macklin Biochemical Technology Co., Ltd. (Shanghai, China). Ammonium acetate buffer (10 mM, pH 6.86) and phosphate buffer (0.1 M, pH 6.8) were obtained from Applygen Technologies Inc. (Beijing, China) LC-MS-grade formic acid was obtained from Fisher-Scientific (Fair Lawn, NJ, USA) whereas LC-MS-grade acetonitrile and methanol were purchased from Merck (Darmstadt, Germany). The distilled water was obtained from Watsons. 

The roots and rhizomes of *panax ginseng* were supplied by the Scientific Research Institute of Beijing Tongrentang Co., Ltd. A total of 14 reference standards (shown in Figure 3), including zingibroside R_1_ (R1), 20(*S*)-ginsenoside Rg_3_ (R2), ginsenoside Rg_6_ (R3), pseudoginsenoside-RT_1_ (R4), chikusetsusaponin IV (R5), chikusetsusaponin Iva (R6), ginsenoside Rd (R7), ginsenoside Ro (R8), ginsenoside F_4_ (R9), ginsenoside Ra_1_ (R10), ginsenoside Ra_2_ (R11), quinquenoside R_1_ (R12), ginsenoside Ra_3_ (R13), and ginsenoside Rc (R14), were purchased from Shanghai Yuanye Bio-Technology Co., Ltd. (Shanghai, China). The purity of all the reference standards was >98%. 

### 3.2. Sample Preparations

The *panax ginseng* was pulverized into powder (just like flour). The powder of *panax ginseng* (15.0 g) was ultrasonically extracted for 30 min with 150 mL 70% methanol at 25 °C. The extracted solution was then filtered through a filter paper. This extraction was repeated twice. The filtrate was combined and evaporated to dryness (3.5 g) using a rotary evaporator at 40 °C. The residue (10 mg) was then dissolved in 4 mL of ammonium acetate buffer (10 mM, pH 6.86) and filtered through a 0.22-μm nylon filter membrane to obtain the sample of affinity ultrafiltration screening.

### 3.3. Screening of α-Glucosidase Inhibitors from Panax Gingseng by Ultrafiltration-ESI-Orbitrap-MS

#### 3.3.1. Affinity Ultrafiltration Screening

The affinity ultrafiltration procedure was performed according to the modified method of Li [28]. α-Glucosidase was dissolved in 10 mM ammonium acetate buffer (pH 6.86) to obtain the α-Glucosidase solution (40 U/mL). A total of 100 μL of 2.5 mg/mL *panax ginseng* sample solution was incubated for 30 min at 37 °C with 100 μL α-Glucosidase (40 U/mL). After incubation, the mixture was filtered through an ultrafiltration centrifugal filter (AMICON ULTRA, 0.5 mL, 10 kDa, Millipore, MA, USA) containing a regenerated cellulose ultrafiltration membrane with a 10,000 MW cut-off at 25 °C. The filter was washed with 250 μL ammonium acetate buffer (pH 6.86) by centrifugal force at 14,000 r/min for 10 min to remove the unbound compounds. This process was then repeated four times. Then, 100 μL of methanol–water (50:50; *v*/*v*, pH 3.30) was added to release the bound ligands, and centrifugation was followed (14,000 r/min for 15 min). The release process was then repeated twice. All of the dissociation solution was combined and evaporated to dryness using a nitrogen-blowing instrument. The residue was re-dissolved in 50 μL methanol–water (50:50; *v*/*v*) for LC-ESI-Orbitrap-MS analysis. The control experiment was carried out with denatured enzyme (in boiling water for 10 min). Each pair of sample and control specimens was prepared in four replicates.

#### 3.3.2. UPLC-ESI-Orbitrap-MS Analysis

The re-dissolved solution was analyzed on a Vanquish™ Flex UPLC system (Thermo Scientific, Waltham, MA, USA) equipped with a binary pump and a thermostated column compartment. Multiple components were separated on a Waters ACQUITY UPLC^®^ BEH C_18_ column (2.1 × 100 mm, 1.7 μm) (Waters, Milford, MA, USA) coupled with an ACQUITY UPLC^®^ BEH C_18_ VanGuard^TM^ Pre-Column (2.1 × 5 mm, 1.7 μm) using mobile phase A (0.1% formic acid/water, *v*/*v*) and mobile phase B (acetonitrile) by the following gradient elution program: 0–7 min, 2–20% B; 7–10 min, 20–25% B; 10–20 min, 25–40% B; 20–25 min, 40–65% B; 25–30 min, 65–95% B. The temperature was set at 35 °C, and the flow rate was 0.3 mL/min. The injection volume was 2 μL.

An Orbitrap Exploris 240 mass spectrometer (Thermo Scientific, Waltham, MA, USA) equipped with a Heated ESI source was used to acquire the mass spectra and negative-ion mode was adopted. The MS parameters were as follows: ion spray voltage: 2500 V, sheath gas: 5.08 L/min, auxiliary gas: 9.37 L/min, ion transfer tube temperature: 320 °C, vaporizer temperature: 350 °C, scan range (*m*/*z*): 150–2000, and collision-energy voltage: 35 V. The full scan was operated at a mass resolution of 60,000 whereas the MS^2^ scan was operated at a mass resolution of 15,000. An internal calibration source, Thermo Scientific EASY-IC^TM^ (Thermo Scientific, Waltham, MA, USA), was adopted to calibrate the entire mass range.

The re-dissolved solution was first analyzed in a full scan mode to minimize signal loss, and then the precursor ions of the selected α-Glucosidase inhibitors were performed targeted MS/MS analysis for structure characterization. 

### 3.4. Data Process

Screening ligands from multiple compounds requires some strategies, and in our experiment, we established an effective data process workflow that mainly included three steps: ①Extracting all compounds in each specimen

To comprehensively screen the ligands of α-Glucosidase from the crude extract of *panax ginseng*, we used a workflow to systematically analyze each compound in the sample and control group. The Compound Discoverer^TM^ software (Thermo Scientific^TM^, version 3.2.0.421) was used for the data process, and all compounds in each specimen were processed by peak alignment and peak extraction based on our extracting workflow “input files → select spectra → align retention time → detect compounds → group compounds”. In the step of “input files,” all LC/MS data files of sample and control specimens were input, whereas in the step of “select spectra,” the entire run time (0–30 min) of the spectra, as well as the negative-ion mode, was selected for further processing. In the “align retention time” step, the retention times of all LC-MS data files were aligned (mass tolerance: 5 ppm), whereas in the “detect compounds” step, all compounds in LC-MS data files were extracted using component elucidator algorithm (mass tolerance: 5 ppm; intensity tolerance: 30; S/N threshold: 3; minimum peak intensity: 100,000; extracted ions: [M-H]^−^, [M-H+HAc]^−^; minimum element composition: CHO, and maximum element composition: C_90_H_190_O_90_. The same substances detected in different addition methods were grouped by molecular weight (mass tolerance: 5 ppm) as well as retention time (RT tolerance: 0.1 min) across all files in the “group compounds” step. Running this extracting workflow, we obtained the information of each compound existing in each specimen, including molecular weight, retention time, peak area, etc., which were used for further data analysis. 

②Calculating the peak area ration of each compound and *t*-test

Ideally, after specific binding to α-Glucosidase, the peaks of compounds incubated with α-Glucosidase showed higher intensities or bigger peak areas than those of compounds incubated with denatured enzyme, meaning the PAR value for a ligand was >1. In this step, we calculated the PAR value of every compound in each specimen, and then the mean PAR value of each compound was calculated. The significant difference in peak area of each compound between the sample and the control group was determined by a two-tailed *t*-test. Finally, potential α-Glucosidase inhibitors were selected based on a mean PAR > 1 and *p* < 0.05 from four replicates.

③Characterizing the structures of ligands

The precursor ions of the selected potential α-Glucosidase inhibitors were performed targeted MS/MS analysis and the ingredients were characterized according to the MS/MS spectra obtained.

### 3.5. Molecular Docking of α-Glucosidase and Ligands

The 3D coordinate of α-Glucosidase was retrieved from the Protein Data Bank (PDB code: 5ZCB) [29]. The α-Glucosidase structure was pre-processed by removing solvent and adding hydrogen atoms. The 2D structures of ligands were downloaded from PubChem and then their 3D structures were generated in Chemdraw Ultra (version 14.0, http://www.cambridgesoft.com/, accessed on 13 January 2023). Molecular docking was performed with AutoDock vina (version 1.5.6, https://vina.scripps.edu/, accessed on 13 January 2023), and structural cartoons were prepared using PyMOL (version 2.4.0, https://pymol.org/2/, accessed on 13 January 2023).

### 3.6. α-Glucosidase Inhibitory Activity Assay

The α-Glucosidase inhibitory activity assay was performed in 96-well plates according to the modified method of Jiang [30]. Both α-glucosidase and pNPG were dissolved in a 0.1 M phosphate buffer (pH 6.8). Each tested compound was also dissolved in 0.1 M phosphate buffer to give solutions of various concentrations. A total of 40 μL of the tested compound solution was mixed with 40 μL α-Glucosidase solution (0.2 U/mL). After incubation for 5 min at 37 °C, 20 μL of pNPG solution (2 mM), which was used as a substrate, was added and then incubated for 30 min at 37 °C. The amount of released nitrophenyl product was measured on an Epoch 2 microplate spectrophotometer (BioTek) at 405 nm. Controls contained the same reaction mixture, except the same volume of phosphate buffer was added instead of a solution of tested compounds. Acarbose was used as the positive control. The inhibition (%) of the tested ligands on α-Glucosidase was calculated as: (A_a_ − A_b_)/A_a_ × 100%, where A_a_ was the absorbance of the control, and A_b_ was the absorbance of the tested compound.

## 4. Conclusions

In our study, α-Glucosidase inhibitors from *panax ginseng* were systematically selected and characterized using affinity ultrafiltration screening combined with the UPLC-ESI-Orbitrap-MS method, which has the advantages of shortening the experimental period, reducing the workload, and achieving large-scale and high-throughput screening compared to the traditional active ingredient selection method based on multiple extraction and separation. The ligands were selected through our established data process workflow based on systematic analysis of all compounds in the dissociation solution, and as a result, a total of 24 ligands were selected as α-Glucosidase inhibitors, including 14 known ginsenosides and 10 unknown ginsenosides. The α-Glucosidase inhibitor’s activity of ligands with definite structures were verified by molecular docking and in vitro enzyme inhibition assay. For ginsenosides, our study first systematically selected and characterized the α-Glucosidase inhibitors and revealed that inhibiting α-Glucosidase activity probably was another important mechanism for hypoglycemic effect of ginsenosides. In addition, our established data process workflow has two advantages. First, when samples containing many overlapping signals, such as the extract of *panax ginseng*, our data process workflow could avoid false positive/negative results due to each compound in the specimens being analyzed. The second advantage was much more ligands were selected including ingredients with lower or much lower intensities which were easily neglected by visual comparison of chromatographic peak intensity on liquid chromatograms or total ion chromatograms mode. Therefore, our established data process workflow can be used to select the active ligands from other natural products using affinity ultrafiltration screening.

## Figures and Tables

**Figure 1 molecules-28-02069-f001:**
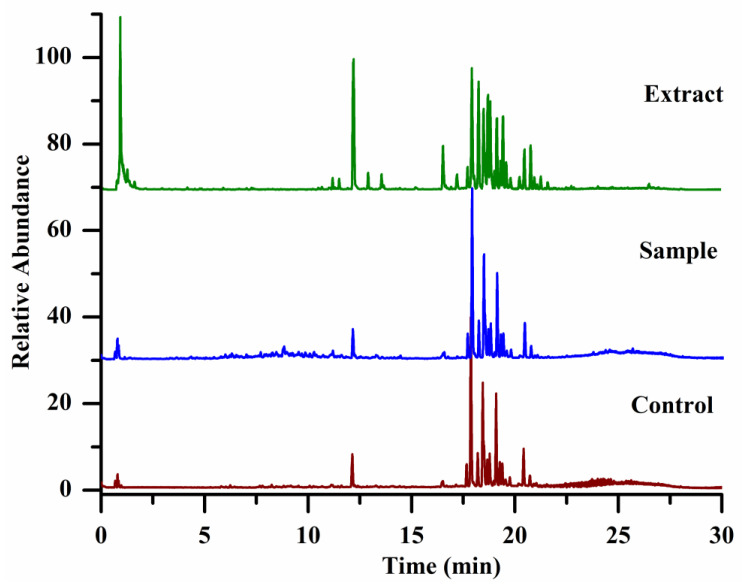
The total ion chromatograms of specimens, the extract of *panax ginseng* (green color); the specimens of the sample group (blue color), and the control group (red color).

**Figure 2 molecules-28-02069-f002:**
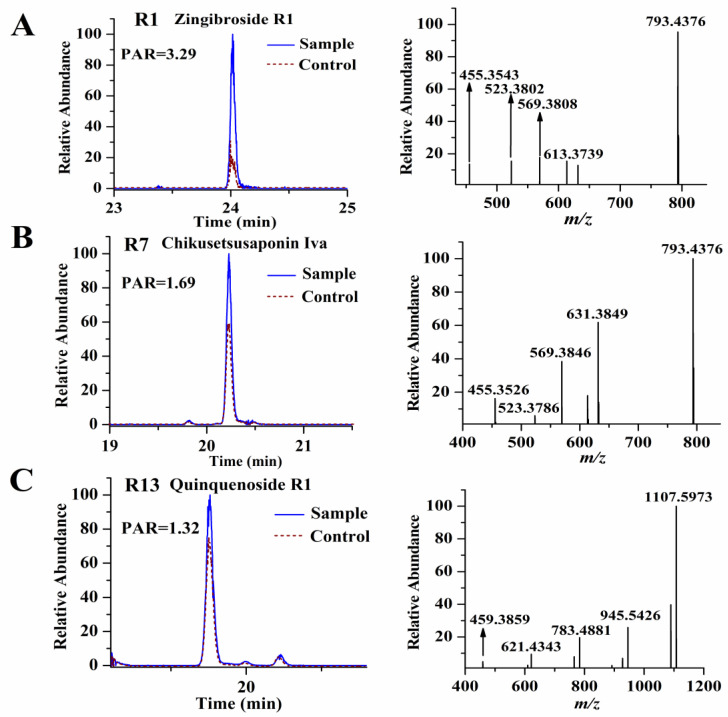
The PAR values of some representative ligands, zingibroside R_1_ (**A**); chikusetsusaponin Iva (**B**); and quinquenoside R_1_ (**C**).

**Figure 3 molecules-28-02069-f003:**
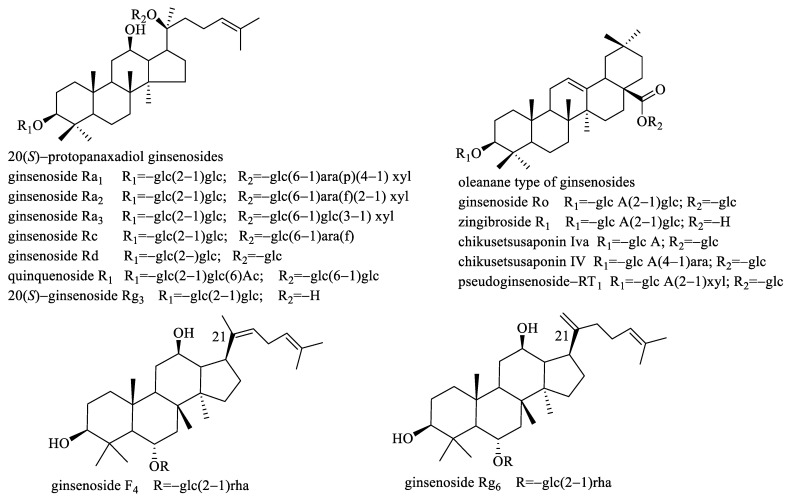
The structures of the 14 (R1–R14) reference standards.

**Table 1 molecules-28-02069-t001:** The α-Glucosidase inhibitors selected from *panax ginseng* extract.

No.	*t*_R_ (min)	Molecular Formula	Calc. MW	PAR Value	Compound Name	Product Ions
Compounds of reference standards commercially available						
R1	24.01	C_42_H_66_O_14_	794.4453 *^a^*	2.69 ± 1.12	zingibroside R_1_	631.3844 [M-H-Glc]^−^, 613.3746 [M-H-Glc-H_2_O]^−^, 569.3849 [M-H-Glc-H_2_O-CO_2_]^−^, 455.3534 [M-H-Glc-Glu A]^−^
R2	24.64	C_42_H_72_O_13_	830.5026 *^b^*	2.02 ± 0.72	20(*S*)-ginsenoside Rg_3_	621.4362 [M-H-Glc]^−^, 459.3860 [M-H-2Glc]^−^
R3	23.34	C_42_H_70_O_12_	812.4924 *^b^*	1.80 ± 0.18	ginsenoside Rg_6_	619.4229 [M-H-Rha]^−^, 457.3707 [M-H-Rha-Glc]^−^
R4	19.44	C_47_H_74_O_18_	926.4856 *^a^*	1.68 ± 0.27	pseudoginsenoside-RT_1_	793.4344 [M-H-Xyl]^−^, 763.4279 [M-H-Glc]^−^, 613.3762 [M-H-Xyl-Glc-H_2_O]^−^, 455.3551 [M-H-Xyl-Glc-Glu A]^−^
R5	19.54	C_47_H_74_O_18_	926.4874 *^a^*	1.59 ± 0.06	chikusetsusaponin IV	793.4389 [M-H-Ara]^−^, 613.3696 [M-H-Ara-Glc-H_2_O]^−^, 455.3551 [M-H-Ara-Glc-Glu A]^−^
R6	20.24	C_42_H_66_O_14_	794.4451 *^a^*	1.57 ± 0.29	chikusetsusaponin Iva	631.3849 [M-H-Glc]^−^, 455.3526 [M-H-Glc-Glu A]^−^
R7	20.44	C_48_H_82_O_18_	992.5542 *^b^*	1.57 ± 0.17	ginsenoside Rd	783.4885 [M-H-Glc]^−^, 621.4365 [M-H-2Glc]^−^, 459.3841 [M-H-3Glc]^−^
R8	18.68	C_48_H_76_O_19_	956.4981 *^a^*	1.52 ± 0.06	ginsenoside Ro	793.4385 [M-H-Glc]^−^, 731.4386 [M-H-Glc-CO_2_-H_2_O]^−^, 613.3731 [M-H-2Glc-H_2_O]^−^, 569.3847 [M-H-2Glc-H_2_O-CO_2_]^−^, 455.3528 [M-H-2Glc-Glu A]^−^
R9	23.63	C_42_H_70_O_12_	812.4924 *^b^*	1.35 ± 0.13	ginsenoside F_4_	619.4215 [M-H-Rha]^−^, 457.3696 [M-H-Rha-Glc]^−^
R10	18.52	C_58_H_98_O_26_	1256.6394 *^b^*	1.33 ± 0.15	ginsenoside Ra_1_	1077.5852 [M-H-Xyl]^−^, 945.5426 [M-H-Xyl-Ara(p)]^−^, 783.4900 [M-H-Xyl-Ara(p)-Glc]^−^, 621.4370 [M-H-Xyl-Ara(p)-2Glc]^−^, 459.3854 [M-H-Xyl-Ara(p)-3Glc]^−^
R11	17.67	C_58_H_98_O_26_	1256.6392 *^b^*	1.25 ± 0.11	ginsenoside Ra_2_	1077.5844 [M-H-Xyl]^−^, 945.5393 [M-H-Xyl-Ara(f)]^−^, 783.4897 [M-H-Xyl-Ara(f)-Glc]^−^, 621.4370 [M-H-Xyl-Ara(f)-2Glc]^−^, 459.3846 [M-H-Xyl-Ara(f)-3Glc]^−^
R12	19.77	C_56_H_94_O_24_	1196.6185 *^b^*	1.23 ± 0.11	quinquenoside R_1_	1107.5973 [M-H-Ac]^−^, 945.5438 [M-H-Ac-Glc]^−^, 783.4898 [M-H-Ac-2Glc]^−^, 621.4370 [M-H-Ac-3Glc]^−^, 459.3844 [M-H-Ac-4Glc]^−^
R13	17.85	C_59_H_100_O_27_	1286.6503 *^b^*	1.15 ± 0.11	ginsenoside Ra_3_	1107.5931 [M-H-Glc]^−^, 945.5460 [M-H-Glc-Xyl]^−^, 783.4923 [M-H-2Glc-Xyl]^−^, 621.4380 [M-H-3Glc-Xyl]^−^, 459.3846 [M-H-4Glc-Xyl]^−^
R14	18.45	C_53_H_90_O_22_	1124.5962 *^b^*	1.14 ± 0.10	ginsenoside Rc	945.5475 [M-H-Ara(f)]^−^, 783.4896 [M-H-Ara(f)-Glc]^−^, 621.4403 [M-H-Ara(f)-2Glc]^−^, 459.3858 [M-H-Ara(f)-3Glc]^−^
Compounds of reference standards commercially unavailable						
R15	17.50	C_48_H_76_O_19_	956.4981 *^a^*	3.81 ± 0.38	ginsenoside Ro isomer	793.4376 [M-H-Glc]^−^, 731.4376 [M-H-Glc-CO_2_-H_2_O]^−^, 613.3737 [M-H-2Glc-H_2_O]^−^, 569.3847 [M-H-2Glc-H_2_O-CO_2_]^−^, 455.3534 [M-H-2Glc-Glu A]^−^
R16	17.92	C_58_H_98_O_26_	1210.6340 *^a^*	2.38 ± 0.16	ginsenoside Ra_1_ isomer/ginsenoside Ra_2_ isomer	1077.5844 [M-H-Xyl]^−^, 945.5478 [M-H-Xyl-Ara]^−^, 783.4885 [M-H-Xyl-Ara-Glc]^−^, 621.4399 [M-H-Xyl-Ara-2Glc]^−^, 459.3831 [M-H-Xyl-Ara-3Glc]^−^
R17	23.11	C_50_H_84_O_19_	1034.5664 *^b^*	1.64 ± 0.13	acetyl-ginsenoside Rd	945.5446 [M-H-Ac]^−^, 783.4888 [M-H-Ac-Glc]^−^, 621.4320 [M-H-Ac-2Glc]^−^, 459.3841 [M-H-Ac-3Glc]^−^
R18	17.97	C_48_H_76_O_19_	956.4982 *^a^*	1.55 ± 0.07	ginsenoside Ro isomer	793.4373 [M-H-Glc]^−^, 731.4371 [M-H-Glc-CO_2_-H_2_O]^−^, 613.3723 [M-H-2Glc-H_2_O]^−^, 569.3849 [M-H-2Glc-H_2_O-CO_2_]^−^, 455.3546 [M-H-2Glc-Glu A]^−^
R19	20.44	C_52_H_86_O_19_	1060.5427 *^b^*	1.51 ± 0.23	(E)-but-2-enoyl ginsenoside Rd	945.5330 [M-H-(*E*)-but-2-enoyl]^−^, 783.4881 [M-H-(*E*)-but-2-enoyl-Glc]^−^, 621.4363 [M-H-(*E*)-but-2-enoyl-2Glc]^−^, 459.3841 [M-H-(*E*)-but-2-enoyl-3Glc]^−^,
R20	19.69	C_48_H_80_O_18_	990.5405 *^b^*	1.39 ± 0.24	dehydrated-protopanaxatriol + 3Glc	781.4766 [M-H-Glc]^−^, 619.4214 [M-H-2Glc]^−^, 457.3699 [M-H-3Glc]^−^
R21	20.74	C_50_H_84_O_19_	988.5601 *^a^*	1.39 ± 0.16	acetyl-ginsenoside Rd	945.5449 [M-H-Ac]^−^, 783.4902 [M-H-Ac-Glc]^−^, 621.4320 [M-H-Ac-2Glc]^−^, 459.3843 [M-H-Ac-3Glc]^−^
R22	19.50	C_58_H_98_O_26_	1210.6350 *^a^*	1.33 ± 0.20	ginsenoside Ra_1_ isomer/ginsenoside Ra_2_ isomer	1077.5856 [M-H-Xyl]^−^, 945.5452 [M-H-Xyl-Ara]^−^, 783.4890 [M-H-Xyl-Ara-Glc]^−^, 621.4329 [M-H-Xyl-Ara-2Glc]^−^, 459.3865 [M-H-Xyl-Ara-3Glc]^−^
R23	17.39	C_54_H_90_O_23_	1152.5928 *^b^*	1.31 ± 0.19	dehydrated-protopanaxatriol + 4Glc	943.5316 [M-H-Glc]^−^, 781.4628 [M-H-2Glc]^−^, 763.4622 [M-H-2Glc-H_2_O]^−^, 619.4229 [M-H-3Glc]^−^, 601.4104 [M-H-3Glc-H_2_O]^−^, 457.3698 [M-H-4Glc]^−^
R24	18.98	C_56_H_94_O_24_	1150.6135 *^a^*	1.31 ± 0.09	quinquenoside R_1_ isomer	1107.5918 [M-H-Ac]^−^, 945.5487 [M-H-Ac-Glc]^−^, 783.4916 [M-H-Ac-2Glc]^−^, 621.4385 [M-H-Ac-3Glc]^−^, 459.3836 [M-H-Ac-4Glc]^−^

*^a^* Calc. molecular weight of [M]; *^b^* Calc. molecular weight of [M + HAc].

**Table 2 molecules-28-02069-t002:** The affinity of ligands for α-Glucosidase and their inhibitory activities on α-Glucosidase.

No.	Compound Name	Affinity (kcal/mol)	IC_50_ (mM)	Inhibitory (%)
R1	zingibroside R_1_	−8.1	3.61	-
R2	20(*S*)-ginsenoside Rg_3_	−8.2	-	UT *^c^*
R3	ginsenoside Rg_6_	−7.8	-	27.35% *^b^*
R4	pseudoginsenoside-RT_1_	−7.8	39.30	-
R5	chikusetsusaponin IV	−8.1	-	16.20% *^b^*
R6	chikusetsusaponin Iva	−7.8	17.33	-
R7	ginsenoside Rd	−7.7	-	UT *^c^*
R8	ginsenoside Ro	−7.8	-	20.23% *^b^*
R9	ginsenoside F_4_	−8.4	22.13	-
R10	ginsenoside Ra_1_	−8.7	-	16.36% *^b^*
R11	ginsenoside Ra_2_	−8.6	-	29.54% *^b^*
R12	quinquenoside R_1_	−7.2	-	25.50% *^a^*
R13	ginsenoside Ra_3_	−9.0	-	17.09% *^a^*
R14	ginsenoside Rc	−7.9	36.83	-
	acarbose	−7.1	5.25	-

*^a^* When the concentration of the single reference standard was 24 mM, the inhibitory rate was determined. *^b^* When the concentration of the single reference standard was 40 mM, the inhibitory rate was determined. UT *^c^* Untested because of the reference standard was insoluble in water.

## Data Availability

The data presented in this study are available on request from the first author.

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
