# Peer review of "Screening of Potential α-Glucosidase Inhibitors from the Roots and Rhizomes of Panax Ginseng by Affinity Ultrafiltration Screening Coupled with UPLC-ESI-Orbitrap-MS Method"

_molecules, 2023, doi:10.3390/molecules28052069_

Round 1
Reviewer 1 Report
Review of manuscript entitled "Screening of potential α-Glucosidase inhibitors from the roots and rhizomes of panax ginseng by affinity selection coupled with UPLC-ESI-Orbitrap-MS method"
The reviewer found this manuscript well organized, written and discussed. The topic is interesting and in the scope of the journal.
I found the manuscript quite novel, the Orbitrap-MS technique used is a new tool in this area. The figures are well presented. The results are well demonstrated. All in all, it is a good manuscript. It has several novelties to others. I have nothing against its publication.
Author Response
Dear reviewer,
Thank you very much for your approval of our article.
Reviewer 2 Report
This manuscript describes an α-Glucosidase inhibitors screening from Panax ginseng by an affinity-based method combined with comprehensive data analysis using UPLC-ESI-Orbitrap-MS, molecular docking, and inhibitory activity assay. It is the first time that ginsenosides were systematically studied on the inhibition of α-Glucosidase. Here are some questions and problems as follows:
1. In the title, abstract, keywords and many places of the text such as lines 35, 71, 272, the concept, ‘affinity selection’, was mentioned, as well as ‘affinity screening’ which was mentioned in lines 34, 133, 269, 389. These two terminologies indicate a similar meaning. It is better to unify them in the manuscript.
2. Page 1 lines 26-34: please improve the writing of the abstract.
3. Affinity selection/screening contains several methods like ultrafiltration, cellular membrane chromatography, and magnetic beads-based ligand fishing. The method that was applied in this study is ultrafiltration. Some expressions in the manuscript about the screening method used in the study need to be narrowed.
4. Line 71, ‘the large-scale, high-throughput filtering.’ should be ‘high-throughput screening’.
5. In the introduction, please consider adding the reference with doi: 10.1016/j.chroma.2020.461740. Rapid Screening α-Glucosidase Inhibitors from Natural Products by At-Line Nanofractionation with Parallel Mass Spectrometry and Bioactivity Assessment
6. Lines 279-281, the results of four times of washing using ammonium acetate buffer should be shown to demonstrate that all unbound compounds were eliminated.
7. For the ‘new data process work flow’, some expressions should be more accurate:
a. Line 315, the Compound DiscovererTM software is developed by the company Thermo Scientific.
b. Lines 373-376, ‘The ligands were selected not by directly comparison of the peak intensity of LC or LC-MS chromatograms of the sample with the control specimens but through our established new data process workflow based on systematically analysis of all compounds in the dissociation solution.’.
However: Line 137-139, ‘From Table 1, we found that the PAR values of the selected were all >1, meaning the intensities of these compounds in the sample group were higher than those in corresponding control group.’. Does this not directly compare the peak intensity then?
Generally, peak areas, peak intensity and retention times are the key parameters for comparison and analysis in affinity-based screening coupled with LC-MS. Normally, many other softwares of LC-MS systems possess functions like peak area and retention time calculation. What is special in the software for the data process? What is new for the data process work flow?
c. Line 84, 125, 126, by the new data process work flow, how to recognize potential inhibitors with lower intensity when they are covered by others? How to calculate the PAR values when peaks contain many overlapping signals and when a peak is covered by others?
d. Lines 384, 385, what causes the false positive/negative results in the method and why can the new work flow avoid them?
8. Some mistakes on grammar and terminology:
a. Line 48-56, ‘For example, ginsenoside Rg1, Rg3, F2, compound K and Rh2, through the regulation of sodium-glucose cotransporters 1 (SGLT1) gene expression to effectively reduce intestinal glucose uptake’. The sentence lacks a predicate verb.
b. Line 50, ‘ginsenoside Rg3 reduced blood glucose and increased plasma glucagon-like peptide-1(GLP-1) and plasma insulin’. ‘reduced’ should be ‘reduces’.
c. Line 60, ‘However, except the above…’. ‘except’ should be ‘besides’.
d. The ‘50’ of ‘IC50’ in the manuscript should be subscript.
e. Figure 2 needs some modification. The numbers are too big. The three chromatograms should be clearly indicated according to the time axis.
In conclusion, it is a good work with complete design and substantial data. However, the reviewer feels that the manuscript exaggerates the innovative character and the function of the new data process work flow. After the authors answer the questions and revise the inaccurate expressions, the manuscript will better describe this application of ultrafiltration to screen inhibitors with comprehensive data analysis.
Reviewer 3 Report
1. Abstract is too wordy.
2. Corresponding author is not labeled in the author list.
3. There are spelling and layout mistakes, e.g. L93, setection>>selection; L104, literal in wrong lines; etc.
4. What is PAR value, the full name should be provided in the first time appear.
5. Table 2, the footnote labeled wrong.
6. The references are too old and should be updated, recent publishing can be cited, e.g. Food Funct., 2022, 13, 2545-2558; Frontiers in Nutrition, 2022, 9, 854882; etc.
Round 2
Reviewer 3 Report
The manuscript was not carefully revised according to the comments。